# The Role of Novel (Tobacco) Products on Tobacco Control in Italy

**DOI:** 10.3390/ijerph18041895

**Published:** 2021-02-16

**Authors:** Silvano Gallus, Elisa Borroni, Anna Odone, Piet A. van den Brandt, Giuseppe Gorini, Lorenzo Spizzichino, Roberta Pacifici, Alessandra Lugo

**Affiliations:** 1Department of Environmental Health Sciences, Istituto di Ricerche Farmacologiche Mario Negri IRCCS, 20156 Milan, Italy; elisa.borroni@marionegri.it (E.B.); alessandra.lugo@marionegri.it (A.L.); 2Department of Public Health, Experimental and Forensic Medicine, University of Pavia, 27100 Pavia, Italy; anna.odone@unipv.it; 3School of Medicine, University Vita-Salute San Raffaele, 20132 Milan, Italy; 4Department of Epidemiology, CAPHRI-School for Public Health and Primary Care, Maastricht University Medical Centre, 6211 LK Maastricht, The Netherlands; pa.vandenbrandt@maastrichtuniversity.nl; 5Department of Epidemiology, GROW-School for Oncology and Developmental Biology, Maastricht University Medical Centre, 6211 LK Maastricht, The Netherlands; 6Oncologic Network, Prevention and Research Institute (ISPRO), 50139 Florence, Italy; g.gorini@ispro.toscana.it; 7Italian Ministry of Health, Center for Disease Prevention and Control, 00144 Rome, Italy; l.spizzichino@sanita.it; 8National Centre on Addiction and Doping, Istituto Superiore di Sanità, 00161 Rome, Italy; roberta.pacifici@iss.it

**Keywords:** smoking prevalence, e-cigarette, heated tobacco products, heat-not-burn tobacco products, harm reduction, Italy

## Abstract

In Italy, electronic cigarettes have spread since 2010 and heated tobacco products (HTP) since 2016. We investigated their public health consequences on conventional cigarette smoking, taking advantage of a series of cross-sectional studies annually conducted between 2001 and 2019 in Italy. Every year, the sample, including around 3000 individuals, was representative of the general Italian population aged ≥15 years. In Italy, smoking prevalence steadily declined from 29.1% in 2001 to 20.6% in 2013, then increased to 22.0% in 2019. In 2017–2019, current electronic cigarette users were 2.1% and in 2019 current HTP users were 1.1%. Among 498 ever electronic cigarette users, 23.2% started or re-started smoking and 15.7% quit smoking after electronic cigarette use; of 49 ever HTP users, 19.1% started or re-started smoking combusted cigarettes and 14.6% quit smoking after HTP use. The availability of novel products in Italy resulted in a halt of the decreasing trend in smoking prevalence. For the first time, we observed an increase of Italians inhaling nicotine, concurrently with the spread of novel (tobacco) products. More importantly, the use of novel products appears to increase—rather than decrease—the likelihood of smoking conventional cigarettes. Considering this evidence, we see no argument to justify the huge fiscal and regulatory benefits these products continue to have, at least in Italy.

## 1. Introduction

Tobacco smoking is the main cause of preventable mortality worldwide: every year more than 8 million deaths are globally attributed to tobacco smoking [1]. In high-income countries, the increased awareness of the harmful effects of tobacco smoking encouraged policymakers to follow the articles outlined in the World Health Organization Framework Convention on Tobacco Control (WHO-FCTC) [2] and adopt stricter and stricter tobacco regulations [3,4,5]. This resulted in a dramatic fall in tobacco sales and consumption over the last few decades [6]. Also in Italy, smoking prevalence was consistently decreasing from 1957 and over the subsequent five decades [7,8]. 

More recently, some novel products, which generate an aerosol containing nicotine or no nicotine, have been introduced into the market. These include electronic cigarettes (since 2010 in Italy) and heated tobacco products (HTP). IQOS, the first HTP by Philip Morris, was launched first in Milan in December 2014, then sold in the whole country from December 2015 and practically spread from 2016 [9]. Electronic cigarettes’ liquids without nicotine (electronic non-nicotine delivery systems, ENNDS [10]) were also commercialized, although their use is limited [11]. All these new products were aggressively promoted by the (tobacco) industry, with the claim that they were less harmful than conventional cigarettes [1,12]. HTPs also circumvented the WHO-FCTC by claiming to be smoke-free and at reduced risk products compared to conventional cigarettes. However, decision n. 22 of the eighth session of the Conference of the Parties (COP8) stated that HTPs meet the definition of tobacco products under FCTC, thus the full range of policy and regulatory measures contained in the WHO-FCTC apply to HTPs [13]. Due to the alleged—but never confirmed—reduced harm, these new products obtained fiscal and regulatory benefits compared to combusted cigarettes in most high-income countries [9,14]. Consequently, these products have gained rapid popularity worldwide [15]. Moreover, likeable flavors and non-regulated appealing advertisements, also on social media, contributed to the spread of electronic cigarettes, attracting in particular the youngest generations [16,17]. Electronic cigarettes and HTPs substantially spread also in Italy—awareness of electronic cigarettes rapidly increased in just a few years from the launch of this novel product [18] and regular electronic cigarette users increased from 0.4% in 2014–2015 to 1.8% in 2016–2017 [19]. Among tobacco products, the market share of IQOS, the first HTP by Philip Morris, increased from 0.01% in 2015 to 0.11% in 2016 up to 0.67% in 2017 [9]. Whereas the safety consequences of these products are still largely unknown [1], more and more concerns have been raised by independent research on public health consequences [1,19,20,21]. In particular: (i) electronic cigarette use has been found to increase—rather than decrease—tobacco smoking in the general adult population [19] and has been associated with tobacco smoking initiation among adolescents [1,22,23,24,25]; (ii) also never smokers or ex-smokers having quit since several years are attracted to these new products [21]; (iii) novel product users are more frequently dual users consuming the novel product where smoking is forbidden [26]; (iv) the role of electronic cigarette on smoking cessation is unclear and uncertain [1] and, as HTPs are tobacco products, the conversion from conventional cigarettes to HTPs should not be considered cessation [1]; and (v) electronic cigarettes increase the risk of renormalization of smoking in society [1,27,28,29]. Moreover, a survey conducted in 12 European countries recently showed how a large proportion of ex-smokers using HTPs did not quit smoking switching to HTPs but they firstly quit and then relapsed to the use of tobacco [30]. In this landscape, it is clear that novel (tobacco) products may have an unfavorable public health effect rather than being harm reduction tools [31,32].

The aim of the present study is to provide updated data on the trends of the prevalence of nicotine consumers in Italy before and after the spread of novel products and to compare the demographic and socio-economic characteristics of conventional cigarette smokers with those of novel product users.

## 2. Materials and Methods

Since 1957 and annually since 2001, the Mario Negri Institute, in collaboration with the Italian National Institute of Health, conducts cross-sectional studies on smoking. Every year, DOXA, the Italian branch of the Worldwide Independent Network/Gallup International Association (WIN/GIA), interviews a sample of around 3000 subjects, representative of the general Italian population aged 15 years and over, in terms of sex, age, geographic area and socio-economic characteristics [8]. For the present study, we considered data from all the surveys conducted by DOXA from 1957 to 2019, with a focus on the surveys conducted in 2017–2019, including a total sample of 9428 individuals (4533 men and 4895 women).

Survey participants were selected through a representative multistage sampling. The first stage involved the selection of municipalities (110 municipalities in 2017, 119 in 2018 and 114 in 2019) in all the 20 Italian regions, based on the region and municipality size. In the second stage, for each municipality, an adequate number of electoral wards was randomly extracted, so that the more or less affluent areas of the municipality were represented in the right proportions. In the third stage, individuals were randomly sampled from electoral rolls, within strata defined by sex and age group. Adolescents aged 15–17 years, who were not included in the electoral lists, were randomly selected by a ‘quota’ method based on the sex and age proportions among them. A statistical weight was generated for each subject to ensure the representativeness of the Italian population aged 15 years or more.

Ad hoc trained interviewers conducted face-to-face surveys using a structured questionnaire in the context of a computer-assisted personal in-house interview (CAPI). Besides general information on socio-demographic characteristics, detailed information was collected on smoking status (never, ex- and current smokers) and other smoking variables. Ever smokers (current and ex-smokers) were participants who had smoked 100 or more cigarettes in their lifetime. Ex-smokers were participants who had quit smoking since at least one year and current smokers were individuals continuing smoking or having stopped since less than one year.

Since 2014, data was also collected on electronic cigarette use. Participants were asked to answer the following question: “Do you use electronic cigarettes or other electronic devices for vaping (disposable or pre-filled or refillable cartridges with liquid), even only occasionally?” (1) Yes, occasionally; (2) Yes, regularly; (3) I used it in the past; (4) No. In 2019, participants were also asked to report their use of HTPs, answering the following question: “Do you use heated tobacco products, like glo or IQOS?” (1) Yes, occasionally; (2) Yes, regularly; (3) I used it in the past; (4) No. Current users of either electronic cigarette or HTPs were defined as those who answered the first or the second items, past users as those answering the third item and never users as those answering the fourth item. Ever users of either electronic cigarette or HTPs were defined as those who were either current or past users (i.e., answering the first, the second or the third item). In 2014–2015 and 2017–2019 electronic cigarette users were asked to report the type of liquid used (with nicotine or without nicotine). We defined participants as “currently using nicotine-containing products,” whether they were either current smokers or current electronic cigarette users consuming nicotine liquids (i.e., excluding users of exclusively non-nicotine liquids) or HTPs users. 

We defined dual users as current cigarette smokers also using novel (tobacco) products (i.e., either electronic cigarette or HTPs, or both).

Level of education was categorized on the basis of participants’ degree into low (up to middle school diploma), intermediate (high school) and high (university). Geographic area was categorized as northern (8 Italian regions), central (4 regions) and southern Italy (8 regions, including islands).

We used official three-month legal sales data on the amount of HTPs and other tobacco products sold in Italy between 2016 (first quarter) and 2019 (last quarter) to compute the market share of HTP tobacco over that period. These data were obtained by the Italian Ministry of Finance [9]. Official legal sales data do not include electronic cigarettes since in Italy they are not classified as tobacco product.

### Statistical Analyses

We carried out a joinpoint regression analysis on the prevalence of current smokers in the overall population and among the young (15–24 years), using the “Joinpoint Trend Analysis Software” developed by the “Surveillance, Epidemiology and End Results Program” of the National Cancer Institute (Bethesda, MD, USA) [33]. “Joinpoints” were identified as time point(s) when a change in the linear slope (on a log scale) of the temporal trend occurred, by testing from zero up to a maximum of four joinpoints for the overall population (23 data points between 1957 and 2019) and of three joinpoints for the young (19 data points between 2001 and 2019). As a summary measure, we also estimated the annual percent change (APC) for each identified linear segment. We derived odds ratios (OR) and corresponding 95% confidence intervals (CI), for current cigarette smokers vs. non-smokers, current electronic cigarette users vs. non-users and current HTP users vs. non-users, through unconditional multiple logistic regression models after adjustment for sex, age, level of education, geographic area and survey year. Statistical analyses were conducted with SAS version 9.4 statistical package (SAS Institute, Cary, NC, USA). All the analyses considered statistical weights to reassure the representativeness of our sample in terms of age, sex, area of residence and socio-economic characteristics. The level of statistical significance was set to a 2-sided *p*-value < 0.05.

## 3. Results

The prevalence of current smokers from 1957 to 2019 among the population aged ≥15 years is shown in Figure 1. In both sexes combined, smoking prevalence steadily decreased from 35.4% in 1957 to 29.1% in 2001 (APC 1957–2001: −0.5%; *p* < 0.05), to 20.6% in 2013 (APC 2001–2013: −2.8%; *p* < 0.05), then restarted increasing and was 22.3% in 2017, 23.3% in 2018 and 22.0% in 2019 (APC 2013–2019: +1.7%; *p* = 0.1).

Figure 2 shows the trends in current smoking prevalence, overall and by sex, in population aged 15–24 years. This prevalence decreased from 34.1% in 2001 to 20.8% in 2019 (APC 2001–2019: −3.2%; *p* < 0.05).

Among Italian adults, the prevalence of current users of nicotine-containing products (i.e., users of any product, including nicotine-containing electronic cigarette—64.1% of all electronic cigarette users—and excluding users of exclusively non-nicotine liquids—35.9% of all users) was 22.2% in 2010–2011, decreased to 20.7% in 2012–2013 and increased subsequently to 21.0% in 2014–2015, to 21.5% in 2016–2017 and to 22.7% in 2018–2019 (Table 1). In 2019, 75% of current electronic cigarette users and 81% of HTP users were dual users (also consuming conventional cigarettes).

In 2017–2019, 22.5% of survey participants were current smokers (26.5% among men and 18.8% among women), 12.5% were ex-smokers and 65.0% were never smokers. In the same period, 2.1% were current, 3.2% past and 94.7% never electronic cigarette users. In 2019, 1.1% were current, 0.5% past and 98.4% never HTP users.

The HTP sold increased from 82 tons in 2016, to 519 tons in 2017, 1522 tons in 2018 up to 3296 tons in 2019. Of all the tobacco products, the corresponding market share was 0.11% in 2016, 0.67% in 2017, 1.99% in 2018 and 4.33% in 2019 (Figure 3). 

Among 498 ever electronic cigarette users, 23.2% reported to have started or re-started smoking as a consequence of their electronic cigarette use, 32.3% did not change their habits, 22.7% decreased their number of cigarettes smoked, 1.4% increased their number of cigarettes smoked, 15.7% quit smoking and 2.9% did not smoke cigarettes before their electronic cigarette use and continued avoiding smoking. Among 49 ever HTP users, 19.1% started or re-started smoking combusted cigarettes, 35.6% did not change their habits, 23.8% decreased their number of cigarettes smoked, 2.1% increased their number of cigarettes smoked, 14.6% quit smoking and 3.3% did not smoke cigarettes before their HTP use and continued avoiding smoking (data not shown).

Table 2 shows the ORs for current cigarette smoking, electronic cigarette use and HTP use according to selected demographic and socio-economic characteristics. Women were less frequently cigarette smokers than men (OR was 0.66; 95% CI: 0.60–0.73). Smoking prevalence was highest among the young and middle-aged participants (compared with ≥65years, OR for <25 was 1.75; 95% CI: 1.42–2.16, OR for 25–44 was 3.23; 95% CI: 2.75–3.80, OR for 45–64 was 2.78, 95% CI: 2.38–3.24) and decreased with increasing level of education (*p* for trend <0.001). Smoking was less frequent in central compared to northern Italian regions (OR was 0.81; 95% CI: 0.70–0.92). No significant relationship has been observed between smoking prevalence and survey year. Current use of electronic cigarettes was less frequent among women (OR was 0.69; 95% CI: 0.52–0.93). Electronic cigarette use increased with decreasing age (*p* for trend <0.001) and with increasing level of education (*p* for trend <0.001). Electronic cigarette use was more frequent in southern compared with northern Italian regions (OR was 1.51; 95% CI: 1.10–2.06) and was less frequent in 2019 compared with 2017 (OR was 0.68; 95% CI: 0.48–0.97). HTP use increased with decreasing age (*p* for trend = 0.029). No significant relationship has been observed with sex, level of education and geographic area.

## 4. Discussion

The inverse trend in smoking prevalence observed in Italy since 1957, particularly favorable between 2001 and 2013, disappeared in the last quinquennium. No acceleration in the decreasing trend was also observed among the young. These findings are in broad agreement with other Italian data from national and international surveillance systems, which observed a declining trend in smoking prevalence until 2012/2013 and a plateau thereafter [34,35]. According to the Eurobarometer, a similar pattern was observed in Spain and Germany but not in other countries that recently enforced strict tobacco control measures, as France and the UK [34]. Our findings empirically confirm our worries of several years ago [18]: in Italy, the availability of novel products (i.e., electronic cigarettes and HTPs) instead of accelerating the process towards a tobacco endgame, provided a detrimental impact on tobacco control. In fact, due to the availability of these novel (tobacco) products, the number of Italians inhaling nicotine (thus using conventional cigarette, electronic cigarettes or HTPs) is increasing for the first time over the last 6 decades. This has also been shown in other studies based on Italian adolescents [36]. One of the reasons why electronic cigarettes and HTPs on tobacco control may have unfavorable effects is that these new products could renormalize nicotine and tobacco product use [1,27,28,29]. 

Although we observed that among both electronic cigarette and HTP users the number of subjects decreasing the number of cigarettes smoked per day is higher than the number of subjects increasing their smoking intensity, we also found that those (re)starting smoking after having used the corresponding novel product outnumber those who stop smoking after having used the novel product. Our findings confirm evidences observed in previous Italian studies [19] and emphasize therefore that, from a public health perspective, novel (tobacco) products have an unfavorable net effect.

Whereas the prevalence of electronic cigarette users decreased in 2019, sales of HTPs markedly increased. One possible explanation is that a part of electronic cigarette users might have recently switched to use HTPs. Today HTPs (mainly IQOS of Philip Morris, whereas glo of British American Tobacco has a marginal role in Italy), representing more than 4% of total tobacco market share, are the third most sold tobacco product after manufactured cigarettes (85%) and roll your own (RYO) tobacco (7%) [9].

We also confirm that novel products more frequently attract the young and individuals with higher socio-economic level. In particular, after adjustment for several covariates including age group, the prevalence of cigarette smokers decreased, whereas prevalence of electronic cigarette users increased, with increasing level of education, taken as a proxy of socio-economic status. It is unlikely that these subjects are part of the small subgroup of the population of hardcore smokers who could really benefit from a harm reduction strategy. On the contrary, it seems that novel products are attracting a new slice of users, confirming findings observed among Italian adolescents [36].

The present study has some limitations inherent to its cross-sectional design. It will be important to confirm our findings with the conduction of large longitudinal studies allowing researchers to evaluate changes in the smoking habits of electronic cigarette and HTP users. The sample size of each annual survey is relatively limited and may be inadequate to observe annual differences for relatively uncommon habits as current electronic cigarette and HTP use. Strengths of the study include the availability of the same survey tool with the same methodology and standardized questions on tobacco smoking, electronic cigarettes and HTPs, annually used for the last two decades.

## 5. Conclusions

In conclusion, our findings from Italy show: (i) unfavorable trends in adult smoking prevalence concurrently with the spread of novel products, (ii) increasing trends of nicotine use for the first time over the last 6 decades, (iii) the majority of novel (tobacco) product users are dual users continuing smoking conventional cigarettes; (iv) the use of novel products increases—rather than decreases—the likelihood of smoking conventional cigarettes; and (v) novel (tobacco) products frequently attract and are used from slices of population who would unlikely benefit of a harm reduction strategy. We therefore confirm what the WHO and most researchers, independent by conflicts of interest, are warning since several years: besides safety issues, electronic cigarettes and HTPs raise public health concerns that let these products be a gateway towards smoking consumption and a threat for tobacco control [1,18,19,37,38]. We therefore concur with the WHO to strongly discourage, at least in Italy, the use of electronic cigarettes and HTPs always, also as an alternative to conventional cigarettes [1]. Today, there is no argument to justify the huge fiscal and regulatory benefits these products continue to have, at least in Italy.

## Figures and Tables

**Figure 1 ijerph-18-01895-f001:**
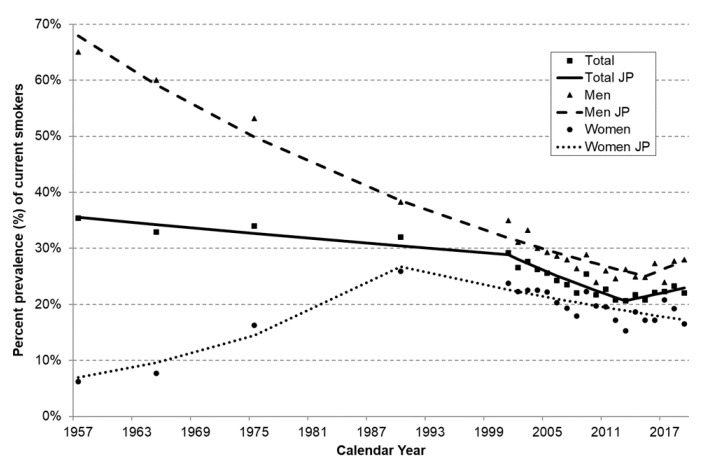
Trends in the prevalence of current smokers among Italian adults aged ≥15 years, overall and by sex. Italy, 1957–2019. APC: Annual Percent Change; CI: confidence interval; JP: Joinpoint.Squares, triangles and dots represent observed prevalence of current smokers overall and among men and women, respectively. Solid, dashed and dotted lines represent the predicted values obtained from the joinpoint regression models overall and among men and women, respectively. Total: 1957–2001: APC = −0.5% (95% CI: −0.7; −0.2); 2001–2013: APC = −2.8% (95% CI: −3.6; −1.9); 2013–2019: APC = +1.7% (95% CI: −0.5; 4). Men: 1957–2015: APC = −1.7% (95% CI: −1.8; −1.6); 2015–2019: APC = +2.6% (95% CI: −4.1; 9.8). Women: 1957–1990: APC = +4.2% (95% CI: 1.1; 7.4); 1990–2019: APC = −1.5% (95% CI: −2.2; −0.8).

**Figure 2 ijerph-18-01895-f002:**
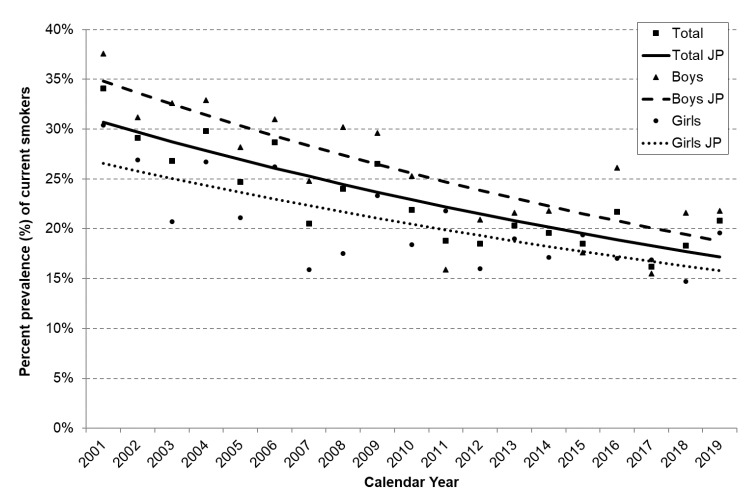
Trends in the prevalence of current smokers among Italian young adults aged 15–24 years, overall and by sex. Italy, 2001–2019. APC: Annual Percent Change; CI: confidence interval; JP: Joinpoint Squares, triangles and dots represent observed prevalence of current smokers overall and among boys and girls, respectively. Solid, dashed and dotted lines represent the predicted values obtained from the joinpoint regression models overall and among boys and girls, respectively. Total: 2001–2019: APC = −3.2% (95% CI: −4.1; −2.2). Boys: 2001—2019: APC = −3.4% (95% CI: −4.5; −2.2). Girls: 2001–2019: APC = −2.9% (95% CI: −4.1; −1.6).

**Figure 3 ijerph-18-01895-f003:**
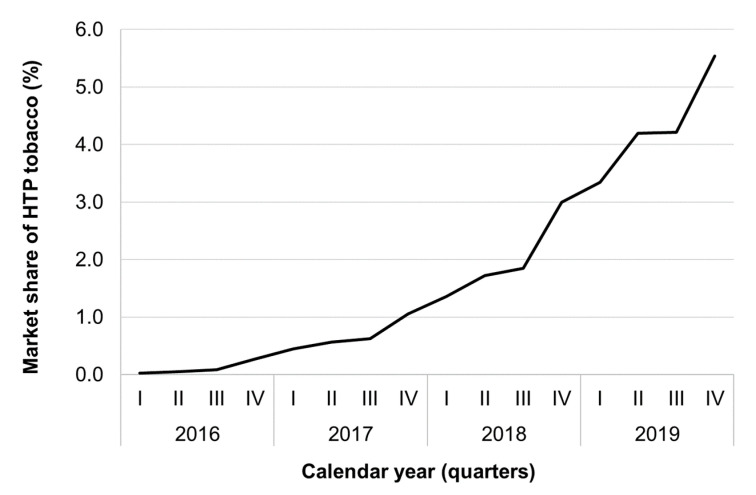
Quarter market share of heated tobacco products (HTP) tobacco (%), Italy, 2016–2019.

**Table 1 ijerph-18-01895-t001:** Trends in the prevalence of Italian adults aged ≥15 years currently using nicotine-containing products, overall and by selected tobacco or nicotine product. Italy, 2010–2019.

Current Use (%)					
	2010–2011	2012–2013	2014–2015	2016–2017 *	2018–2019
Cigarettes only	22.2	20.7	20.2	20.7	21.1
Electronic cigarettes only			0.1	0.2	0.3
Cigarettes AND electronic cigarettes			0.7	0.6	0.8
HTPs only					0.1
Cigarettes AND HTPs					0.2
Electronic cigarettes AND HTPs					0.0
Cigarettes AND electronic cigarettes AND HTPs					0.2
Total (users of nicotine-containing products)	22.2	20.7	21.0	21.5	22.7

Abbreviation: HTP: heated tobacco product * In 2016 no information on type of liquid used (nicotine vs. non-nicotine liquid) for electronic cigarette users was available. Thus, for 2016 all the electronic cigarette users have been considered.

**Table 2 ijerph-18-01895-t002:** Prevalence of current smokers among 9428 Italian participants aged ≥15 years, overall and by selected demographic and socio-economic characteristics. Corresponding odds ratios* and 95% confidence intervals. Italy, 2017–2019.

	N° (2017–2019)	Current Cigarette Smokers		Current Electronic Cigarette Users		N° (2019)	Current HTP Users #	
		%	OR (95% CI)	%	OR (95% CI)		%	OR (95% CI)
Total	9428	22.5	-	2.1		3120	1.1	
Sex								
Men	4533	26.5	1 ^	2.5	1 ^	1501	1.5	1 ^
Women	4895	18.8	**0.66 (0.60–0.73)**	1.7	**0.69 (0.52–0.93)**	1619	0.7	0.54 (0.27–1.11)
Age								
<25 years	1064	18.4	**1.75 (1.42–2.16)**	2.5	**2.14 (1.16–3.94)**	352	1.3	5.00 (0.95–26.15)
25–44 years	2821	28.8	**3.23 (2.75–3.80)**	2.6	**2.28 (1.35–3.86)**	929	1.8	**6.69 (1.53–29.15)**
45–64 years	3136	26.5	**2.78 (2.38–3.24)**	2.4	**2.29 (1.37–3.82)**	1039	1.0	3.68 (0.84–16.21)
≥65 years	2407	12.0	1 ^	0.8	1 ^	800	0.3	1 ^
*p* for trend			**<0.001**		**<0.001**			**0.029**
Education level								
Low	3183	20.1	1 ^	1.2	1 ^	1028	0.8	1 ^
Intermediate	4655	25.7	0.94 (0.83–1.06)	2.7	**1.88 (1.27–2.80)**	1579	1.5	1.09 (0.46–2.59)
High	1590	18.3	**0.59 (0.50–0.70)**	2.2	1.52 (0.93–2.48)	513	0.4	0.31 (0.07–1.40)
*p* for trend			**<0.001**		**<0.001**			0.066
Geographic area								
Northern Italy	4332	23.1	1 ^	1.9	1 ^	1430	1.3	1 ^
Central Italy	1890	19.9	**0.81 (0.70–0.92)**	1.7	0.92 (0.61–1.40)	624	0.6	0.47 (0.16–1.44)
Southern Italy	3206	23.3	0.97 (0.87–1.09)	2.7	**1.51 (1.10–2.06)**	1066	1.1	0.88 (0.41–1.86)
Survey year								
2017	3086	22.3	1 ^	2.5	1 ^	-	-	-
2018	3222	23.3	1.08 (0.96–1.22)	2.1	0.85 (0.61–1.18)	-	-	-
2019	3120	22.0	0.99 (0.88–1.12)	1.7	**0.68 (0.48–0.97)**	3120	1.1	-
*p* for trend			0.446		0.114			

Abbreviations: CI: confidence interval; HTP: heated tobacco products; OR: odds ratio. * ORs were estimated using unconditional multiple logistic regression models after adjustment for sex, age, level of education, geographic area and survey year. Estimates in bold are statistically significant at 0.05 level. # Analyses limited to 2019. Total numbers of survey participants overall and in each strata of the population. ^ Reference category.

## Data Availability

Data are available upon reasonable request to the corresponding author.

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
