# Peer review of "The Role of Novel (Tobacco) Products on Tobacco Control in Italy"

_ijerph, 2021, doi:10.3390/ijerph18041895_

Round 1

Reviewer 1 Report

Overall this paper is helpful in describing the tobacco use patterns in Italy, particularly with the inclusion of e-cigarettes and HTPs.  However, a few concerns present themselves upon review.  First, the following paragraph in the conclusions does not seem well supported:

Whereas the prevalence of electronic cigarette users decreased in 2019, sales of HTPs 227
markedly increased. This suggests that a relevant part of electronic cigarette users recently 228
switched to use HTPs. Today HTPs (mainly IQOS of Philip Morris, whereas Glo of British 229
American Tobacco has a marginal role in Italy), representing more than 4% of total tobacco 230
market share, are the third most sold tobacco product after manufactured cigarettes (85%) 231
and roll your own (RYO) tobacco (7%).

--It appears to be an oversimplification of the data to say that e-cig users became HTP users based on the data presented.  Also, is there evidence or a reference available for the claim that HTP is the third most sold product?  Given the low percentages of HTP use in this sample it seems unlikely to be true, particularly given that e-cigarettes have a higher percent of use.

I am also unclear as to how dual use is accounted for an understood in the context of the findings presented?  How does dual use impact the interpretation of HTP and ENDS use and use patterns compared to cigarettes?

Finally, given the finding that HTP and e-cig use is a higher socioeconomic activity, and cigarette use has decreased while novel product use has increased, greater discussion would be helpful to understand the role of SES in these changes in use patterns and concerns in increasing use of novel products over time.

Reviewer 2 Report

Thank you very much for the opportunity to review the manuscript: "The Role of Novel (Tobacco) Products on Tobacco Control in Italy", by Silvano Gallus et al.

In a cross-sectional study, the authors collected smoking related data from ~3000 subjects from selected regions of Italy, with the aim to get a representative view on prevalence of the whole country. Odds ratios were calculated by unconditional multiple logistic regression models for various pairs of smoking status. Also, a jointpoint regression analysis was performed, to analyse the temporal trends of smoking prevalence.

The authors reported, that smoking prevalence in Italy steadily declined from 1957 until ~2013, but rather increased between 2013 and 2019. This recent rise in smoking prevalence started at the time of introduction of new tobacco products, namely HTP and electronic cigarettes.

The authors investigated the impact of these new products on smoking prevalence and conclude that they their use is increasing the likelihood of smoking conventional cigarettes. Thus, the introduction of new tobacco products is providing a possible explanation for the worrisome change in prevalence timetrends.

The authors of this study already reported about smoking prevalence in Italy for the years 2015-2016, including a table showing the development since 2007 where the increase since 2013 is already slightly visible. The authors demonstrate in this paper, that the positive time-trend in Italy has been reversed in direction since 2013 and offer a plausible and very likely explanation.

The manuscript is written in good English and may just require a minor check. Methods are appropriate and statistical power is sufficient to support the conclusions.

There is no significant point of criticism from my side, just one point that caught my attention: There is a WHO Trends Report on tobacco use from 2019, showing data from several European countries, including Italy. This report also covers the period 2013-2019. In this report, the increase of prevalence in Italy since 2013 is not visible. The database, for 2013-2019, in the WHO-report is given to be mainly national for Italy, with contributions from EHIS only for 2014. Can the authors shortly comment, what could be the reason for the discrepancy/difference? Did the authors contribute to the database used for the WHO report? However, this question is rather motivated by interest and not a point of criticism.

Reviewer 3 Report

I would like to thank the Editor for the opportunity of reviewing this interesting manuscript.

Before my review, and due to the relevance of this topic and for the sake of transparency with the authors, I would like to say that I do not have any conflicts of interest associated, I do not support the tobacco harm reduction strategy and I do support the tobacco endgame.

This is a properly conceptualized, methodologically sound and well-written manuscript. Authors make use of readily available cross-sectional data in Italy to assess the association between the spread of novel products and the prevalence of cigarette smoking, although longitudinal designs are necessary (as the authors correctly address in the limitations) to define a causal association.

Although this is a manuscript based at the national level, I think it can be of interest for a broader audience.

My main concern with the manuscript is the way authors support some of their conclusions based on the design of the study and the available results.

I provide some specific questions/comments/suggestions below:

Introduction:

Perhaps it would be of interest to elaborate on how HTP circumvent the FCTC.

Line 45: “which generate an aerosol containing nicotine”. Perhaps it would be good to mention ENNDS.

Line 52-53: are fiscal and regulatory benefits the only reason behind the current popularity of new products in youngsters? In my opinion, the appealing forms of these devices and the (obscene) flavors used, for example, may merit a line or two here.

Materials and Methods:

Line 81: “annually” is redundant with “annually” in line 80.

Lines 104-107: is there any rationale behind the classification of this smoking status? Personally, I find them a bit away from some of the standards (at least those I know of) (e.g. NHIS).

Line 117: “currently using nicotine”. Do authors have information on the type of delivery system used (ENDS/ENNDS)? Perhaps not every ecig user is a nicotine consumer.

Statistical analyses:

I believe it would be better to firstly address the joinpoint regression, and the logistic regression afterwards, to keep the order followed for the results.

How is the variable “level of education” defined? What does “high”, “intermediate” and “low” refer to?

Which are the categories of the variable “geographic area”?

Is there any reason behind setting the maximum number of joinpoints to three (e.g. number of observations)?

Please indicate if significance level was set at 0.05.

Results:

Lines 141-143: I would encourage the authors to provide confidence intervals here. Also, there are two % missing for APCs. Authors may provide specific p-values, if available.

Figure 2: while the legend says “boys” and “girls”, “men” and “women” are used in the text.

Table 1: heading says “addicted to nicotine”. Nicotine addiction is a disorder falling under F17 ICD-10. I would encourage the authors to modify this wording.

Also on Table 1, if the interviews are carried out annually, why does the heading correspond to two years for each column? Are % an average?

Table 2: Authors state that the model is adjusted for geographic area, but coefficients for this variable are not provided. Please also indicate the n for the column of smokers and users.

Line 165: please see comment above on “current nicotine users”.

Line 176-178 (and Figure 3): perhaps this is just an opinion, but I believe this excerpt removes attention from the main (and highly interesting results) of this manuscript, although I find the comments to these data interesting in Discussion.

Line 181 and 185: “ever electronic cigarette users” and “ever HTP users” should be defined in Materials and Methods. If you have data on former users, perhaps it would be interesting to see the % of these who are current smokers.

Discussion:

Line 210-217: Is this trend also observed in other European/EU countries? Perhaps it would be good to provide results on this to further support the authors’ findings.

Line 219: I think “on tobacco control” does not fit there.

Line 222-224: I absolutely agree with the authors that this is a major finding, but do the authors believe that the “full picture” (i.e. “the unfavourable net effect on tobacco control” of these products, of which I am also convinced) is depicted with this finding? I would encourage the authors to elaborate on other findings of their manuscript (e.g., the % of Italians increasing and decreasing the number of cigarettes smoked) and beyond to provide this “full picture”, especially due to the wide scope of readers of the journal.

Conclusion iii) should say “the majority of novel (tobacco) product users…”

Abstract:

Line 25-26: the n for ever electronic cigarette users and HTP users should be added.

Lines 27-31: I also believe that the trend is worrying since it seems to have changed, and I agree with you on the implication in line 31-33, but I think these statements should be moderated. The wording sounds as a causal association, which is not sustained by the current design.
